# CYP20-3 deglutathionylates 2-CysPRX A and suppresses peroxide detoxification during heat stress

Wenshan Liu*, Izailda Barbosa dos Santos*, Anna Moye, Sang-Wook Park

In plants, growth-defense trade-offs occur because of limited resources, which demand prioritization towards either of them depending on various external and internal factors. However, very little is known about molecular mechanisms underlying their occurrence. Here, we describe that cyclophilin 20-3 (CYP20-3), a 12-*oxo*-phytodienoic acid (OPDA)–binding protein, crisscrosses stress responses with light-dependent electron reactions, which fine-tunes activities of key enzymes in plastid sulfur assimilations and photosynthesis. Under stressed states, OPDA, accumulates in the chloroplasts, binds and stimulates CYP20-3 to convey electrons towards serine acetyltransferase 1 (SAT1) and 2-Cys peroxiredoxin A (2CPA). The latter is a thiol-based peroxidase, protecting and optimizing photosynthesis by reducing its toxic byproducts (e.g., $H_2O_2$). Reduction of 2CPA then inactivates its peroxidase activity, suppressing the peroxide detoxification machinery, whereas the activation of SAT1 promotes thiol synthesis and builds up reduction capacity, which in turn triggers the retrograde regulation of defense gene expressions against abiotic stress. Thus, we conclude that CYP20-3 is a unique metabolic hub conveying resource allocations between plant growth and defense responses (trade-offs), ultimately balancing optimal growth phonotype.

## Introduction

Peroxiredoxins (PRXs) are ubiquitous and the most abundant thiol-based peroxidases capable of reducing a broad range of toxic peroxides in the presence of sufficient electron ($H^+$, $e^-$) donors. The peroxidase cycle starts by their redox-active cysteine (Cys; the peroxidatic Cys, $C_P$) in the catalytic center that is oxidized to sulfenic acid ($C_P$-SOH) by a broad set of peroxide substrates. The $C_P$-SOH residue then reacts either with another Cys of the same or an adjacent subunit, or with another thiol-containing compound, to form an inter- or intramolecular disulfide ($C_P$-S-S-$C_R$), which is later reduced (via a mixed disulfide with a reductant) to reform the thiolate (Perkins et al, 2015; Liebthal et al, 2019).

In Arabidopsis, the nuclear genome encodes two plastid 2-CysPRX (2CP) isoforms (denoted A and B) that play crucial, versatile roles in plant growth and survival, acting as reductase, redox sensor, and chaperone, along with their peroxidase detoxification property in protecting and modulating photosynthetic mechanisms (Muthuramalingam et al, 2009). However, 2CPs are known to typically form an obligatory homodimer as the $C_P$ from one monomer is connected via a disulfide bond to the resolving Cys ($C_R$) located at another monomer. The oxidation of $C_P$ then, in consequence, deactivates the catalytic activity of 2CPs. Thus, 2CP dimers require electron donors such as an NADPH-dependent thioredoxin reductase C (NTRC), thioredoxins (TRXs), and/or cyclophilin 20-3 (CYP20-3), which reduces (activates) them to be able to metabolize the detoxification of a toxic by-product in photosynthesis (i.e., $H_2O_2$), and the activation of Calvin cycle enzymes such as fructose 1,6-bisphosphatase (Dietz et al, 2006; Caporaletti et al, 2007; Laxa et al, 2007; Muthuramalingam et al, 2009; Liebthal et al, 2016). Therefore, deficiency of 2CPs in antisense, and T-DNA insertion mutant plants manifested the increased levels of $H_2O_2$ and carbonylated proteins, while decreasing the quantum yield of PSII and $CO_2$ fixation rates, which together result in growth and developmental inhibition (Baier & Dietz, 1999; Baier et al, 2000; Pulido et al, 2010; Awad et al, 2015).

On the other hand, 2CPs could display an array of oligomeric structures upon cellular positions and conditions, as well as other posttranslational modifications (PTMs; Perkins et al, 2015; Liebthal et al, 2019). Under oxidative stress, 2CPs could be overoxidized and form a homodecameric complex that disables their peroxidase activity, but instead gains a chaperon activity to protect cellular molecules against oxidative damage (Liebthal et al, 2019). However, recent studies with human PrxI (huPrxI) and pea 2CP have argued that 2CPs are rather S-glutathionylated by GSSG during oxidative stress, protecting their quaternary structures to remain as dimers and, as a result, enzymatically inactivated status (Park et al, 2009, 2011; Calderón et al, 2017).

Among several plastid enzymes of which transcripts are coregulated with 2CPs, stromal CYP20-3 is one that appeared to directly interact with them (Muthuramalingam et al, 2009; Cheong et al, 2017). CYP20-3 is a dual enzyme, exerting chaperon (i.e., peptidyl-prolyl *cis–trans* isomerase and PPIase) and reductase activities (Laxa et al, 2007; Park et al, 2013), positioned as a regulatory hub between the light-dependent reaction in photosynthesis and 12-*oxo*-phytodienoic acid (OPDA) signaling (Cheong et al, 2017). OPDA is

Department of Entomology and Plant Pathology, Auburn University, Auburn, AL, USA

Correspondence: swpark@auburn.edu
*Wenshan Liu and Izailda Barbosa dos Santos contributed equally to this work

a primary precursor of (-)-jasmonic acid (JA), able to trigger an autonomous signaling pathway that regulates unique subsets of jasmonate-responsive genes, activating and fine-tuning plant defense responses, as well as growth processes (Böttcher & Pollmann, 2009; Dave & Graham, 2012). Its distinctive activity was first described by the pathoanalyses of a mutant Arabidopsis (opr3) arresting the conversion of OPDA to JA (Stintzi et al, 2001). Wild type (WT)–like resistance of opr3, in contrast to decreased resistance in mutant plants disrupting trienoic fatty acid biosynthesis (fad3/7/8) and the octadecanoid pathway (dde2 and aos), against fungal and insect infections underlined the essential roles of OPDA signaling in plant defense responses in the absence of JA and JA-Ile (Stintzi et al, 2001; Zhang & Turner, 2008; Stotz et al, 2011). Following studies with several mutant plants, suppressing or impairing JA production (e.g., siOPR3, OPR3-RNAi, cts-2/opr3 and acx1) or OPDA signaling (cyp-20-3) further substantiated that OPDA signaling is crucial in basal defense responses against a variety of pathogenic fungi and insects, such as Alternaria brassicicola, Botrytis cinerea, Scierotinia sclerotiarum, Nilaparvata lugens, Manduca sexta, and Bradysia impatiens, as well as seed germination, embryogenesis, and balancing abscisic acid signaling (Dave et al, 2011; Goetz et al, 2012; Park et al, 2013; Bosch et al, 2014; Guo et al, 2014; Scalschi et al, 2015).

Under stressed conditions, OPDA, accumulates in the chloroplasts, binds and promotes CYP20-3 to transfer electrons from the photosystem I (PSI) via TRXs (type-f2 and -x) towards 2CPs (Motohashi et al, 2001; Laxa et al, 2007; Dominguez-Soils et al, 2008; Cheong et al, 2017) or a serine acetyltransferase 1 (SAT1, Dominguez-Soils et al, 2008; Park et al, 2013). Reduction of 2CPs then controls peroxide (photo-oxidant) detoxifications and photosynthetic carbon metabolisms (Dietz et al, 2006; Caporaletti et al, 2007), whereas the activation of SAT1 stimulates the plastid sulfur assimilation, which leads to the production of Cys and thiol metabolites (e.g., glutathione; GSH), and the buildup of cellular reduction potential (Park et al, 2013). The enhanced reduction capacity, in turn, coordinates the expression of a subset of OPDA-responsive genes (ORGs) and general defense regulators (e.g., glutaredoxin 480) in controlling basal and race-specific (local and systemic) resistances and defense responses against various abiotic stresses (Mou et al, 2003; Park et al, 2013).

Collectively, available data suggest that the functional dynamics of 2CP isoforms in conjunction with CYP20-3–dependent OPDA signaling fine-tunes energy inputs into outputs that shape plant growth and defense response ("trade-offs"), programing optimal phenotypes under different ecological conditions. In this context, the present study demonstrates that heat stress (HS) prompts CYP20-3 to temporally limit an antioxidant machinery of GSH-glutathionylated 2CPA (2CPA$^{GS}$) in photosynthesis, while relaying an OPDA signal, which triggers the retrograde regulation of nucleus defense gene (e.g., HSP17.6, HSP70, and CYP18D11) expressions. The HS responses explain a unique molecular mechanism underlying the mode of resource allocations between plant growth and defense responses. Besides, these data also highlight a novel activity of GSH as a functional group of posttranslational modifiers, apart from its antioxidant activity, which determine (i) the quaternary structure and (ii) the cellular activity of enzymes (e.g., 2CP$^{GS}$ isoforms), and (iii) directed their metabolic pathways (i.e., reductant signaling), controlling the interface between plant growth, defense responses, and stress acclimation processes.

# Results

## 2CPA$^{GS}$ and 2CPB$^{GS}$ form discrete quaternary structures

Recently, emerging evidence have elucidated a critical role of redox-mediated PTM in resolving the cellular property and modus operandi of 2CPs (Park et al, 2009, 2011; Calderón et al, 2017). In agreement, our preparatory analyses uncovered that 2CPs, prepared recombinantly in Escherichia coli, uniquely bind a negatively charged tripeptide GSH, a major nonprotein thiol antioxidant in plants (Fig S1). GSH-binding (hereafter, called GSH-glutathionylation) then differentially modulates the conformational states of 2CPs (Fig 1A lanes 1, 2, 5, and 6, and Fig S2), stimulating predominantly the monomerization and peroxidase activity of 2CPA, while decamerizing and accentuating the chaperone activity of 2CPB (Fig S3, Lee et al, 2015).

However, the two plastid 2CPs, sharing a high sequence identity (>96% in amino acids, Fig S4), have been considered to be functionally and structurally redundant, controlling peroxide detoxifications and carbon metabolisms in photosynthesis (Kirchsteiger et al, 2009; Pulido et al, 2010). Thus, to further scrutinize whether the distinctive conformations are an intrinsic property of 2CPs$^{GS}$ and not caused by noncoding amino acids derived from expression vectors, we re-prepared and examined quaternary structures of the "tag-free" version of recombinant 2CPs (called as 2CPs hereafter, Fig 1A). As anticipated, 2CPs and 2CPs$^{GS}$ displayed mostly a similar format of quaternary structures to His-tagged 2CPs and 2CPs$^{GS}$. A notable variance was that 2CPA$^{GS}$ comprised only di- and monomers (Fig 1A lane 4), whereas His-tagged 2CPA$^{GS}$ constituted a tripartite conformation, that is, deca-, di-, and monomers (lane 2). Nevertheless, both 2CPB$^{GS}$ and His-tagged 2CPB$^{GS}$ alike formed icosa- and decameric conformations (lanes 6 and 8), supporting an earlier notion that 2CPA$^{GS}$ and 2CPB$^{GS}$ configure distinctive quaternary states (Figs S2 and 3A) and, in consequence, confer unique cellular functions as a peroxidase (2CPA$^{GS}$, Fig S3B) and a molecular chaperone (2CPB$^{GS}$, Fig S3C), respectively.

## Positions of Val and Ile determine discrete quaternary structures between 2CPA$^{GS}$ and 2CPB$^{GS}$

Mature 2CP sequences differ in seven amino acids (Fig S4), suggesting that those residues are likely responsible for the disparate formation of quaternary structures between 2CPA$^{GS}$ and 2CPB$^{GS}$. To substantiate this hypothesis, we comparatively surveyed the quaternary structures of single and double-mutant 2CPBs of which specific amino acids were replaced with corresponding ones in 2CPA (i.e., E$_{33}$D, Y$_{64}$H/E$_{65}$S, V$_{106}$I/I$_{109}$V, P$_{122}$H, and V$_{157}$I; Fig 1B). Most mutant 2CPBs and 2CPBs$^{GS}$ however exhibited the WT-like quaternary structure, except one (V$_{106}$I/I$_{109}$V). The two nearby amino acids, V$_{106}$ and I$_{109}$, in 2CPB were switched to I$_{106}$ and V$_{109}$, respectively, located at the corresponding positions in 2CPA. This mutant 2CPBV$_{106}$I/I$_{109}$V indeed behaved like 2CPA, unable to form deca- and icosamers but releasing monomers upon GSH-glutathionylation (Fig 1B lane 4 and 10). In comparison, an analogous mutant 2CPAI$_{106}$V/V$_{109}$I exhibited 2CPB-like behavior (Fig 1B lane 14), illuminating a crucial role of Val and IIe, and their specified positions at 106 and 109 in the oligomerization of 2CPB, as well as 2CPB$^{GS}$. A series of hydrophobic connections via V$_{106}$ from one monomer and I$_{109}$ located at another monomer may lead to the decamerization of 2CPB and 2CPB$^{GS}$. Conversely, the reverse positions of I$_{106}$ and V$_{109}$ in 2CPA structurally discommoded the V-I

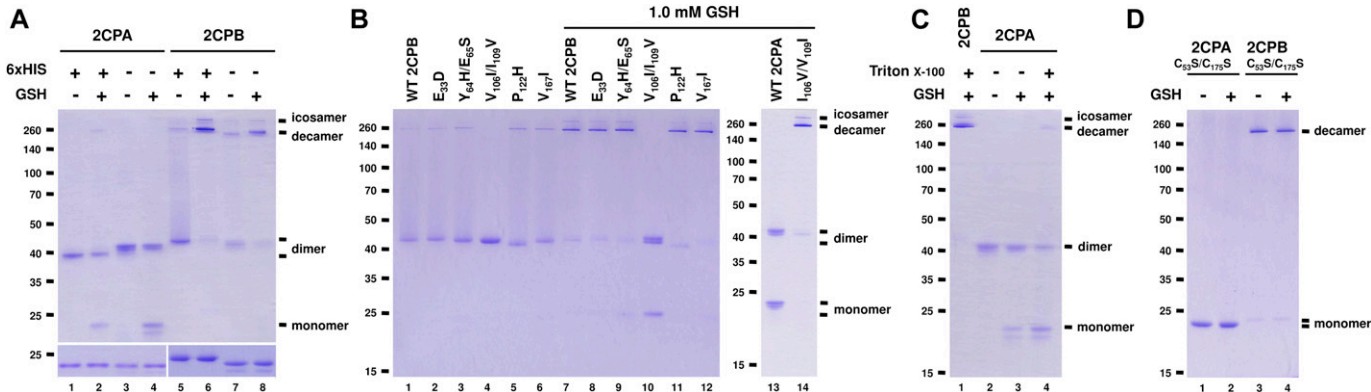

**Figure 1. Val and Ile determine different quaternary structures between 2CPA$^{GS}$ and 2CPB$^{GS}$.**
**(A, B, C, D)** Redox shift visualization of WT and/or mutant 2CPs$^{GS}$. **(A)** His- and nontagged versions of 2CPA or 2CPB (1 $\mu$M) were incubated with/without 1 mM GSH and subjected to nonreducing (upper panel) or reducing (lower panel) SDS/PAGE. **(B)** Mutant 2CPBs (E$_{33}$D, Y$_{63}$H/E$_{65}$S, V$_{106}$I/I$_{109}$V, P$_{112}$H, and V167I; 1 $\mu$M) and mutant 2CPA (I$_{106}$V/ V$_{109}$I; 1 $\mu$M) were incubated with/without 1 mM GSH. **(C)** WT 2CPA or 2CPB (1 $\mu$M) was incubated with/without 1 mM GSH and/or 0.1% (vol/vol) Triton X-100. **(D)** Cys to Ser mutant 2CPs (C$_{53}$S/C$_{175}$S; 1 $\mu$M) were incubated with/without 1 mM GSH. Data information: In (A, B, C, D), recombinant 2CPs were produced in *E. coli* and purified by a nickel column, as described in the Materials and Methods section. Gels were stained with Coomassie Brilliant Blue, and standard molecular weight sizes were indicated in the left of gels. Each lane number was denoted below the gel. In (B, C, D), all proteins were tag-free versions and separated via nonreducing SDS/PAGE. Source data are available for this figure.

interactions and a subsequent decamerization of 2CPA$^{GS}$ (Fig 1C). A partial denaturation (relaxation) of its tertiary structure enabled to form decameric complexes upon the GSH-glutathionylation (Fig 1C lane 4 versus 2 and 3). In addition, the disruption of intradisulfide bonds in C to S mutant 2CPs (C$_{53}$S/C$_{175}$S) casts an action on the V–I interactions which resulted predominantly in decameric 2CPB, in contrast to the monomeric 2CPA produced, regardless of the GSH-glutathionylation (Fig 1D). It is though notable that mutant 2CPBC$_{53}$S/C$_{175}$S, unlike 2CPB$^{GS}$ (Fig 1A lane 7 and 8), did not form icosamers (Fig 1D lanes 3 and 4), indicating that the S–S interactions maybe needed for the icosamerization of 2CPB and 2CPB$^{GS}$, perhaps via the dimerization of decameric 2CP complexes.

### Increased ratios of GSH:GSSG coordinate the GSH-glutathionylation of 2CPA

Next, we investigated whether cellular GSH homeostasis may influence the rate of GSH-glutathionylation of 2CPs. For instance, earlier studies from our and other groups revealed that the activation of plant defense responses via salicylic acid (SA) and OPDA signaling systematically induce GSH synthesis, independent of oxidative stress signaling, and build up cellular reduction potentials (increased GSH-to-GSSG ratio, Mou et al, 2003; Park et al, 2013). In a resting state, the chloroplasts that produce SA and OPDA maintained GSH-to-GSSG ratios of 14:1 (Koffler et al, 2013), and these escalated up to ≥ 28:1 under stress conditions (Park et al, 2013). The enhanced redox capacity then stimulates the GSH-glutathionylation and monomerization of 2CPA$^{GS}$ (Fig 2A lane 3–5), which in turn likely accentuated its peroxidase activity (Fig S3B). 2CPB$^{GS}$ however was unresponsive to the elevated level of GSH and maintained largely deca- and icosamers (Fig 2B). On the contrary, decreased reduction (GSH-to-GSSG ratio) capacity displayed little effect on the GSH-glutathionylation of both 2CPs (Fig 2A and B lanes 6 and 7). The latter further supports a unique and autonomous activity of GSH in activating redox signaling, independent of GSSG-mediated S-glutathionylation and its oxidative signaling (Xiong et al, 2011; Grek et al, 2013), that relays hormone (e.g.,

OPDA; Fig S5) signaling during the stress-responsive activation of defense and acclimation pathways.

### GSH-glutathionylation suppresses the reducing activity of NTRC, TRX, and SRX towards 2CPs$^{GS}$

Previously, several in vitro studies have surveyed the reducing activity of major plastid redox mediators, elucidating that TRX and NTRC can break up symmetrical S–S bridges in 2CP dimers, or that sulfiredoxin (SRX) deglutathionylates a GS–S bond in 2CPs$^{GS}$ (Park et al, 2009; Yoshida & Hisabori, 2016). We hence examined whether and how TRX, NTRC, and/or SRX metabolize the structure and function of 2CP$^{GS}$. Note however that the reaction of TRX and NTRC often required excessive reducing powers (≥500 $\mu$M DTT or NADPH, Yoshida & Hisabori, 2016), ≥300-fold greater than the physiological concentration of NADPH (<1.5 $\mu$M; Maruta et al, 2016), that in consequence caused the nonenzymatic reduction, perhaps deglutathionylation, of 2CPs$^{GS}$ (Fig S6). Therefore, we lowered the level of supplement of DTT (10 $\mu$M) and NADPH (50 $\mu$M) to avoid their direct impacts on the redox state of 2CPs$^{GS}$ (Fig 3A–D). Herein, TRX and NTRC exhibited minimal reductase activity, exhibiting little if any effect on the quaternary structure of 2CPs$^{GS}$. Likewise, SRX did not metabolize nor deglutathionylate 2CPs$^{GS}$ (Fig 3E and F). Though, it was noticeable that the supplement of SRX causes a slight delay in the gel mobility of partial 2CPBs$^{GS}$ (Fig 3F), indicating that SRX may be able to target the S–S bond forming 2CPB$^{GS}$ icosamers (Fig 1D). Indeed, a daily expression rhythm of *SRX* was coregulated with those of *2CP* transcripts (Fig 3G and H). When plants were grown under 12-h light/12-h dark diurnal conditions, the expression of both *2CPA* and *2CPB* was hiked along with *SRX* in the afternoon, supporting a potential activity of SRX towards 2CPs$^{GS}$. However, *TRX* peaked at darkness, whereas *NTRC* was constitutive, projecting its biochemical and biological irrelevance to 2CPs$^{GS}$. Caveat is that 2CPs are highly abundant (~0.6% of the total plastid proteins) and determined to exhibit slow turnover rates (Horling et al, 2003; Dietz et al, 2006). Hence, the co-expression of *2CPs* and *SRX* may not, at once, tie in their physiological and functional interactions.

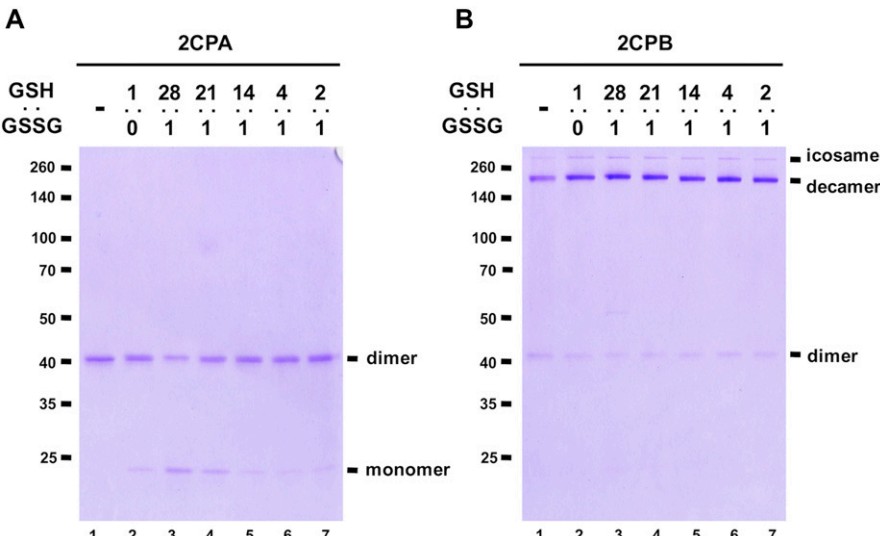

**Figure 2.  Redox potential controls the conformational and functional state of 2CPA[GS].**
**(A, B)** The 1 µM, tag-free recombinant 2CPA (A) and 2CPB (B), incubated with different ratios of 1 mM GSH:GSSH (1: 0, 28:1, 21:1, 14:1, 4:1, and 2:1) were subjected to nonreducing SDS/PAGE and stained with Coomassie Brilliant Blue. Standard molecular weight sizes were indicated in the left of gels.
Source data are available for this figure.

### CYP20-3 deglutathionylates 2CPA[GS] at high temperatures

A series of protein–protein interaction and DNA protection analyses have inferred that CYP20-3 is another electron donor of 2CPs (Laxa et al, 2007; Muthuramalingam et al, 2009; Cheong et al, 2017), but our redox-shifting assays did not detect any reductase activity of CYP20-3 towards 2CPs[GS] (Fig S7). We therefore speculated whether CYP20-3 confers rather PPIase activity and protects 2CPs[GS] from thermal aggregations (at 45°C, Bhuwan et al, 2017). However, 2CPs[GS] turned out to be heat stable (Fig 4A and B lane 2), making us unable to measure the chaperone activity of CYP20-3. We instead noticed at ≥36°C that CYP20-3 becomes able to convert monomeric 2CPAs[GS] to dimers (Fig 4A lane 5 and 6), although it shows little effect on 2CPB[GS] (Fig 4B lanes 5 and 6). When incubated at 42°C, CYP20-3 was able to reduce a mixed GS–S bond (i.e., deglutathionylation, Fig 4C right panel) and led 2CPA[GS] to form an obligatory dimer (Fig 4C left panel). On the other hand, CYP20-3 dissipated icosameric 2CPB[GS] (Fig 4D right panel) proposing either (i) its target-specific deglutathionylation activity to 2CPB[GS] icosamers or (ii) its ability to break a proposed S–S bonds between decameric 2CPBs[GS] (see above in Fig 1D). Nonetheless, most 2CPB[GS] (or 2CPB) remained as decamers, an active molecular chaperone (Fig 4D left panel), disregarding a substantial activity of CYP20-3 towards 2CPB[GS]. Moreover, the high temperature (42°C) caused little change in the catalytic accessibility of 2CPs[GS] to other reductases such as NTRC, TRX, and SRX (Figs S8 and S9). Thus, these observations indicated the specific and decisive roles of CYP20-3 in HS-dependent deglutathionylation of 2CPA[GS].

### CYP20-3 blocks the peroxidase activity of 2CPA[GS] during heat-shock stress

Our data explain that a reduction of 2CPA[GS] frees GSH (deglutathionylation) and makes up 2CPA dimers (Fig 4C). These conversely resemble the $H_2O_2$-mediated oxidation and inactivation of 2CPA (Fig 1A lane 1 and Fig S2B). Indeed, the deglutathionylation of 2CPA[GS] paralleled an attenuation of its peroxidase activity, reducing $H_2O_2$ (Fig 5A), indicating that CYP20-3 conveys a temporal suppression of the $H_2O_2$

detoxification system of 2CPA[GS] during HS. In line with this scenario, $H_2O_2$ levels increased rapidly following HS (peaking at ~4 h post HS; hph) and remained till 12 hph in WT, *2cpb* and *ntrc*, but gradually reduced in *cyp20-3*, or continued increasing up to ~8 hph in *2cpaI*, *2cpaII*, and *2cpa/2cpb* (Fig 5B). Taken together, our data delineate that HS triggers the rapid bursts of $H_2O_2$ (oxidative stress) signaling, while fostering CYP20-3 to temporally limit the peroxidase activity of 2CPA[GS] (i.e., detoxification system), which in fact supports a notion that 2CPA[GS] is a nonessential component in the protection mechanism against oxidative stress (Laxa et al, 2007). Note that the expression of *CYP20-3* and *2CPA* was constitutive regardless of HS (Fig 5C).

The acclimation of HS (i.e., HS responses) is largely characterized by expression of a battery of HS proteins (HSPs), many of which are molecular chaperones involved in correct native folding and/or assembly of other proteins (Finka et al, 2011). This explains the HS-induced accumulation of, already abundant, 2CPB[GS] (Fig 5C) that constitutes a stable, decameric conformation (Fig 1) conferring chaperon activity (Fig S3C). On the other hand, HS responses convey plant defense (OPDA) signaling (Muench et al, 2016) that activates CYP20-3–dependent sulfur assimilation in increasing thiol metabolites, which then builds up cellular reduction potential (Park et al, 2013). The enhanced redox capacity, in turn, coordinates the expression of a subset of ORGs, including *HSP17.6*, *CYP81D11*, and *HSP70* (Figs 5D and S10). Therefore, OPDA-insensitive mutant (*cyp20-3*) hindered the expression of HS-responsive genes (i.e., ORGs; *HSP17.6*, *HSP70*, and *CYP81D11*), whereas the disruption of an NTRC system assisted increased accumulations of *HSP17.6*. These results support the versatile activity of CYP20-3 in OPDA signaling, which conveys the activation of disease resistance against *A. brassicicola* and defense responses to different abiotic stresses such as wounding and HS (Park et al, 2013; Figs 5D and S10).

## Discussion

In this study, we attempted to elucidate the molecular basis of unique functions between 2CPA[GS] and 2CPB[GS] isoforms. Until recently, two 2CPs have been considered to be functionally and

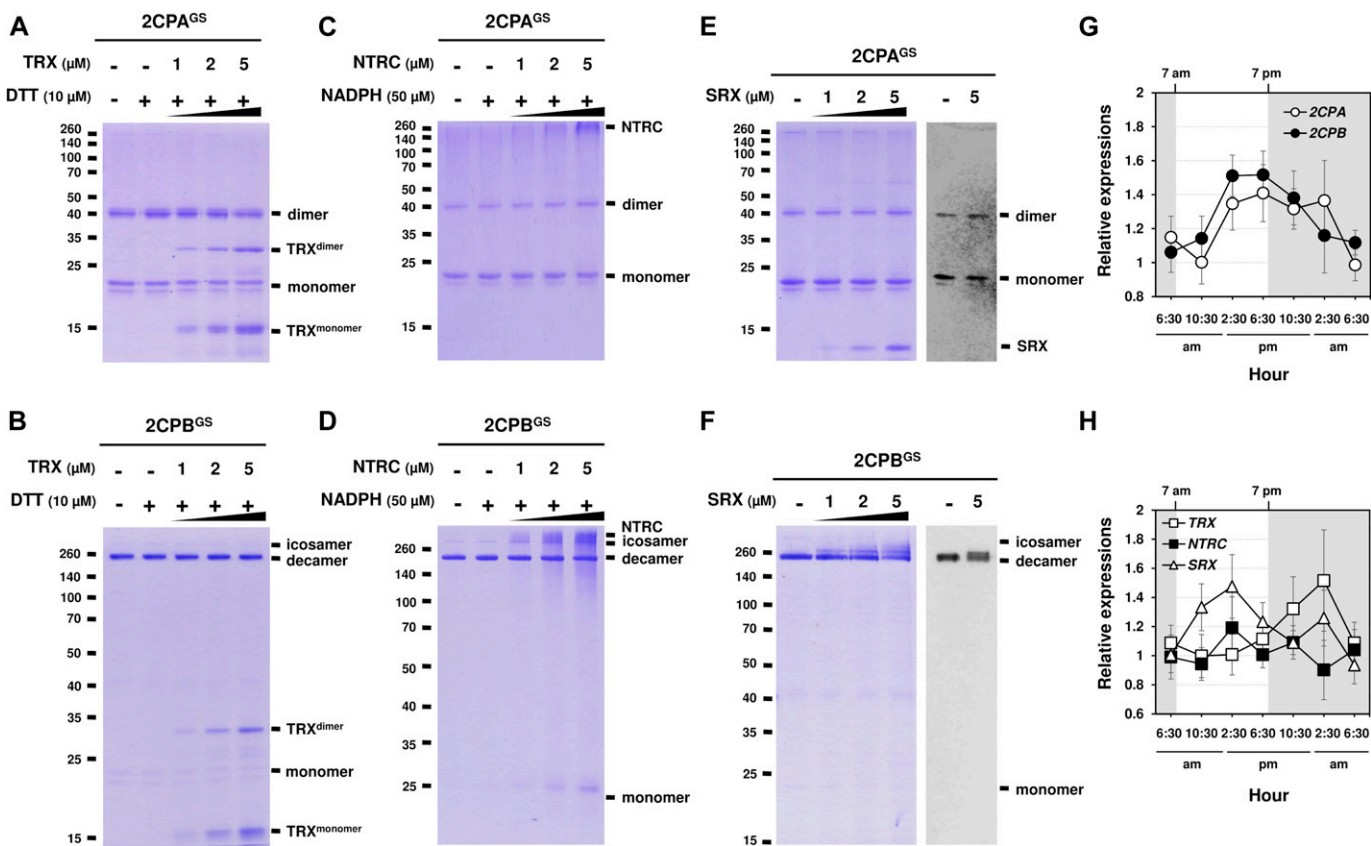

**Figure 3. GSH-glutathionylation arrests the catalytic accessibility of 2CPs[GS] towards major plastid-redox mediators, TRX, NTRC, and SRX.**
**(A, B, C, D, E, F)** Redox shift visualization of 2CPs[GS]. 1 $\mu$M 2CPA[GS] and 2CPB[GS] were incubated with DTT/TRX (10 $\mu$M DTT and 0–5 $\mu$M TRX; A, B), NADPH/NTRC (50 $\mu$M NADPH and 0–5 $\mu$M NTRC; C, D), or SRX (0–5 $\mu$M; E, F). **(E, F, right panel)** 2CPA[GS] (E) and 2CPB[GS] (F), incubated with/without 5 $\mu$M SRX were analyzed by Western blot using a monoclonal anti-GSH antibody. All proteins were tag-free, recombinant versions prepared in *E. coli* BL21 (DE3) and separated by nonreducing SDS/PAGE. Gels were stained with Coomassie Brilliant Blue, and standard molecular weight sizes were indicated in the left of gels. **(G, H)** Transcript quantification by quantitative RT-PCR of *2CPs* (G) and *TRX*, *NTRC*, and *SRX* (H) under diurnal 12-h light/12-h dark conditions. Total RNAs were prepared in every 4 h from the leaves in WT (Col-0) Arabidopsis plants, and values were normalized to the expression of three reference genes, *UBC*, *GAPDH*, and *PP2A* (means ± SD; $n$ = 3).
Source data are available for this figure.

structurally redundant, controlling peroxide detoxifications and carbon metabolisms in photosynthesis (Kirchsteiger et al, 2009; Pulido et al, 2010). A series of biochemical assays however unveiled that 2CPs are an intrinsic target of the GSH-glutathionylation; their protein sulfhydryl groups (PSH) of the resolving Cys ($C_R$) can directly bind and form mixed disulfides with GSH. This PTM then differentially modulates and protects the structure and function of 2CPs against various cellular and ecological constraints, leading to the monomerization and peroxidase activity of 2CPA[GS], while decamerizing and enhancing the chaperone activity of 2CPB[GS].

The key determinant underlying the distinctive quaternary structures between 2CPs[GS] turned out to be two amino acid residues, Val and Ile. Both 2CPs, in fact, contain Val and Ile, but their positions are reversed each other locating at $I_{106}$ and $V_{109}$ in 2CPA, whereas at $V_{106}$ and $I_{109}$ in 2CPB. The latter then fit the tertiary structure of 2CP to allow the V–I interactions and make up the decameric complex of 2CPB and 2CPB[GS]. Val and Ile, aliphatic residues are able to form a network of hydrophobic and van der Waals interactions, which often calibrate and stabilize the binding structure of single protein, multiprotein, and protein–ligand systems (Dill & MacCallum, 2012; Zhu et al, 2016). Indeed, their iso-butyl,

sec-butyl, and iso-propyl analogs provide an enhanced capacity for stabilizing the van der Waals interactions (Kathurai et al, 2016), which in turn assists the preferential folding of proteins and protein clusters (Rose & Wolfenden, 1993). This folding stability perhaps explains greater structural and functional integrity of 2CPB[GS] decamers towards enzymatic and chemical restrictions, than those of monomeric and dimeric 2CPA[GS], and proposes an alternative role in supplying compensatory energy to stress-susceptible interactions and/or the structure versatility within protein complexes.

Besides the V–I interactions, 2CPA's quaternary structures and functions are governed by intra-disulfides and/or the GSH-glutathionylation. Until recently, 2CPA was known to form an obligatory homodimer. The oxidation though deactivates the peroxidase activity of 2CPA and thus requires electron donors such as NTRC, TRXs, and/or CYP20-3, which reduce (activate) dimers to be able to metabolize the removal of toxic peroxides (Laxa et al, 2007; Kirchsteiger et al, 2009; Yoshida & Hisabori, 2016). However, our studies showed that 2CPA is principally activated through binding to GSH upon arrival at the chloroplasts. GSH herein should be targeting the resolving Cys ($C_R^{53}$) and enabling 2CPA[GS] to use the peroxidatic Cys ($C_P^{175}$) for catalytic reactions. Hence, monomeric 2CPAs[GS] likely improve the unit of catalytic efficiency,

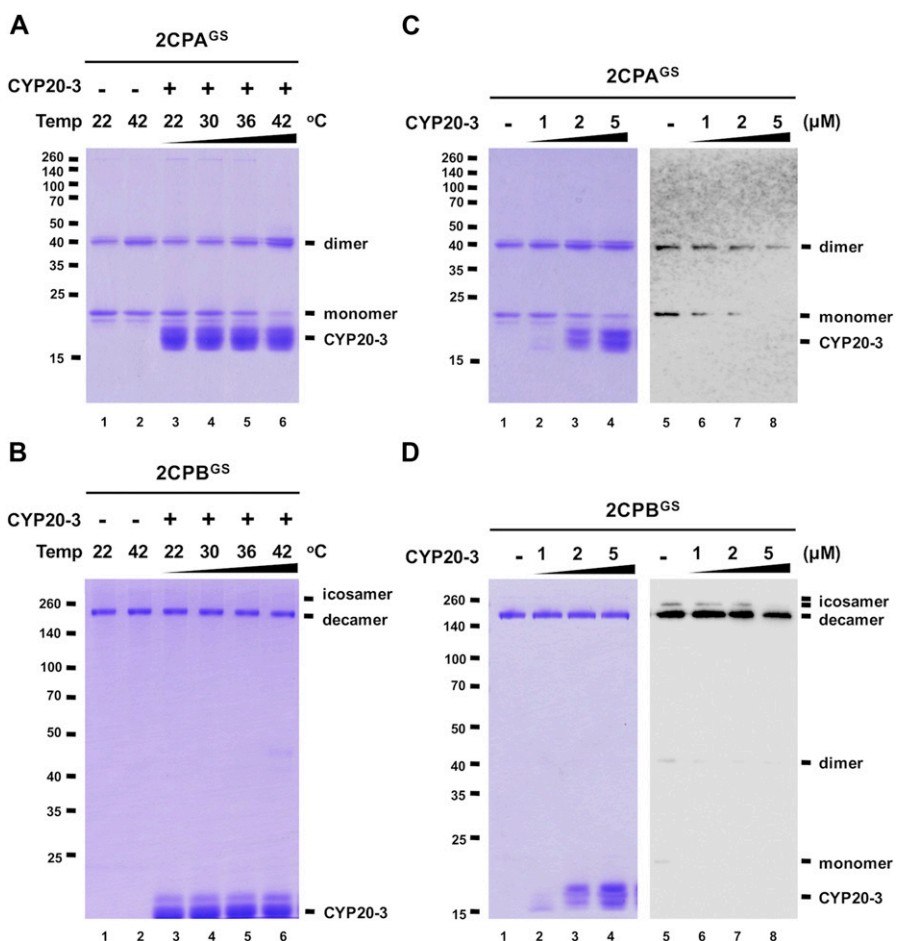

**Figure 4. CYP20-3 deglutathionylates 2CPA^GS at high temperatures.**
**(A, B)** Redox shift visualization of 1.5 μM 2CPA^GS (A) and 2CPB^GS (B) incubated with 5 μM CYP20-3 under increasing temperatures (22, 30, 36, and 42°C) for 10 min were subjected to nonreducing SDS/PAGE. **(C, D)** Redox shift visualization of 1.5 μM 2CPA^GS (C) and 2CPB^GS (D) incubated with various concentrations of CYP20-3 (0–5 μM) at 42°C for 15 min were separated in nonreducing SDS/PAGE (left panel) and probed by Western blot analyses using a monoclonal anti-GSH antibody (right panel). Data information: In (A, B, C, D), all proteins were tag-free, recombinant versions prepared in *E. coli* BL21 (DE3). Gels were stained with Coomassie Brilliant Blue, and standard molecular weight sizes were indicated in the left of gels. Source data are available for this figure.

which is fostered nonenzymatically by the enhanced binding kinetics of GSH via increased cellular reduction capacity (increased GSH-to-GSSG ratios). On the contrary, an enzymatic reduction proceeds the deglutathionylation (inactivation) of 2CPA^GS. The present study demonstrated that temporal HS can foster the enzymatic accessibility of 2CPA^GS towards CYP20-3, resulting in a cleavage of a GS–S bond in 2CPs^GS and a dimerization of 2CPA. The deglutathionylation then inactivates the peroxidase activity of 2CPs^GS, leading to attenuate detoxification mechanisms during HS acclimation processes.

In line with this scenario, earlier studies from our and other groups revealed that the activation of plant defense responses via SA and OPDA signaling systematically induces GSH synthesis independent of oxidative stress signaling (Mou et al, 2003; Park et al, 2013). For instance, OPDA binds and stimulates CYP20-3 to form a complex with SAT1, which triggers the formation of a hetero-oligomeric Cys synthase complex (CSC) with *O*-acetylserine(thiol)lyase B in the chloroplasts. CSC formation then leads to the production of Cys (sulfur assimilation) and subsequently GSH, building up reduction capacity, which in turn activates a subset of ORGs (Park et al, 2013), possibly via fostering a target-specific GSH-glutathionylation that modulates the cellular activity of oxidoreductase cascades (Tada et al, 2008) in controlling retrograde signaling, rapidly adjusting nuclear gene expressions to handle diverse ecological conditions (Mou et al, 2003; Park et al, 2013). Note that our jasmonate quantifications in *cyp20-3* KO mutants (Park et al, 2013) suggest that in a

resting states, CYP20-3 could sequester OPDA and reduce downstream jasmonate productions, but the increased accumulations of OPDA under stress conditions could circumvent the impact of its binding to CYP20-3, exhibiting little difference in JA accumulations between WT and *cyp20-3*, together proposing that OPDA and JA signaling are activated in parallel and/or accumulatively in defense responses.

Recently, emerging evidence has illuminated a unique activity of plant hormone signaling in converting light inputs into outputs that shape the optimal phenotype ("fitness") towards constant environmental challenges (Ballaré, 2014). The cost of resistance, often referred to as a growth versus defense trade-off, has been typically described as a teeter-totter model where for defense to increase, growth must decrease and vice versa. However, very little is known about the molecular mechanisms underlying their occurrence (Huot et al, 2014). The present study locates CYP20-3 as a unique player in controlling the interface between OPDA signaling (defense) and light-dependent redox reactions (growth). When the PSI antenna captures solar energy (in resting states), it prompts a chain reaction of electron transfers that elicits TRX- and NTRC-based redox regulation in controlling energy (sugar) conversion and consumptions, wherein CYP20-3 is positioned to convey electrons from TRXs towards preferentially SAT1. This maintains a basal-level cellular redox homeostasis (Wirtz & Hell, 2006; Dominguez-Soils et al, 2008; Takahashi et al, 2011). By contrast, under stressed conditions, OPDA is accumulated and binds CYP20-3 to stimulate its interactions and

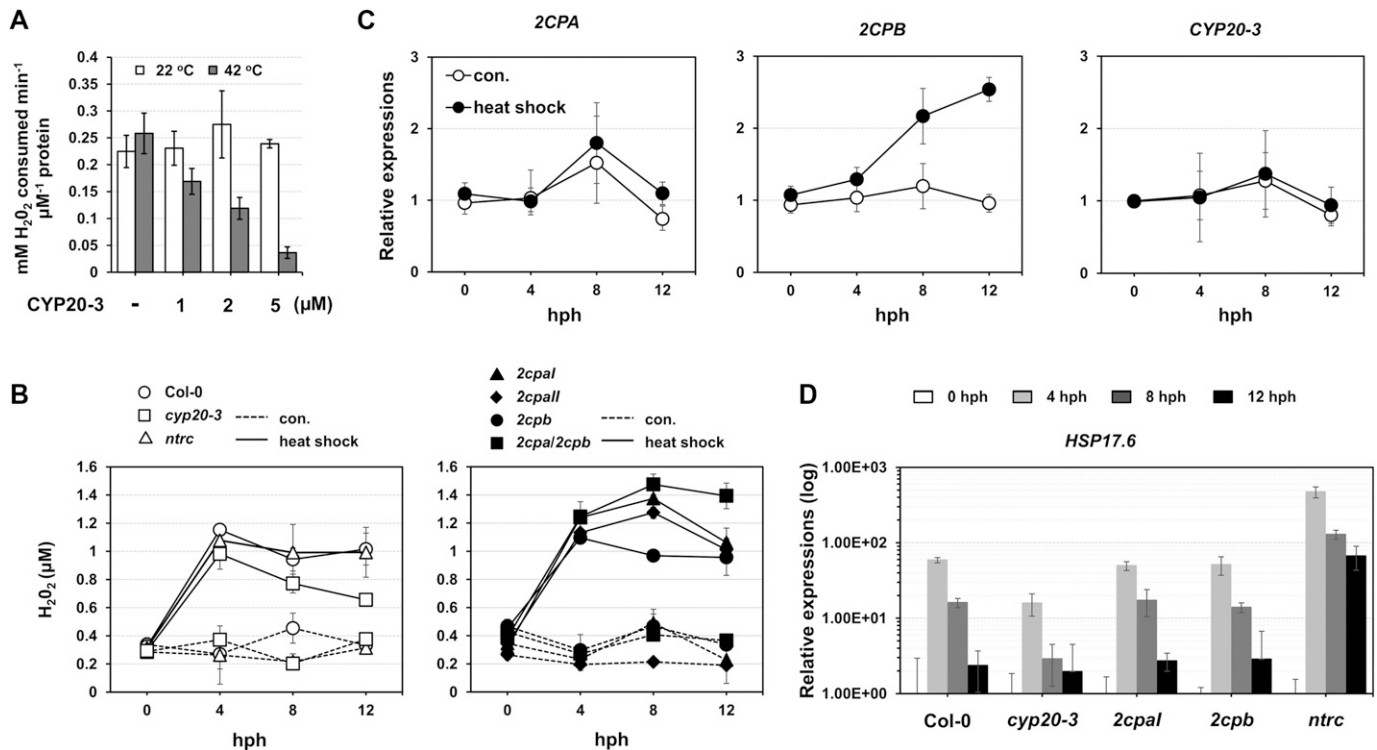

**Figure 5. CYP20-3 suppresses a $H_2O_2$-detoxification system of $2CPA^{GS}$ during heat-shock stress.**
**(A)** Peroxidase activity of $2CPA^{GS}$, upon incubation with various concentrations (0–5 $\mu M$) of CYP20-3 at 22°C (white bars) or 42°C (grey bars) for 15 min, was measured by subsequently incubating with $H_2O_2$ for 10 min. $H_2O_2$ was then quantified using the eFOX method (Cheeseman, 2006). 2CPA and CYP20-3 were tag-free, recombinant versions produced in *E. coli* BL21 (DE3). **(B, C, D)** Time-resolved (0, 4, 8, 12 hph) measurements of $H_2O_2$ (B), as well as RT-PCR quantifications of *2CPs*, *CYP20-3* (C), and *heat shock protein 17.6* (*HSP17.6*; D) in heat-stressed WT (Col-0) and/or mutant (*cyp20-3*, *2cpaI*, *2cpaII*, *2cpb*, *2cpa/2cpb*, and *ntrc*) plants (means ± SD; *n* = 3). Data information: In (C, D), values were normalized to the expression of three reference genes, *UBC*, *GAPDH*, and *PP2A* (means ± SD; *n* = 3). Source data are available for this figure.

electron transfers from TRX-f2 to both SAT1 and $2CPA^{GS}$ (Cheong et al, 2017). Reduction (deglutathionylation) of $2CPA^{GS}$ then inactivates photo-oxidant detoxification and energy biogenesis (Figs 4 and 5), whereas the activation of SAT1 stimulates CSC formation and builds up reduction capacity, which in turn triggers the retrograde regulation of defense gene expression (Park et al, 2013; Cheong et al, 2017). This model sheds new light on (i) a unique interface (CYP20-3) between light and hormone (OPDA) signaling, which (ii) fine-tunes resource (electron) allocations between growth and defense responses (iii) in making instant and appropriate adaptive decisions while being challenged constantly by environmental pressures, maintaining necessary growth and development, and ultimately balancing optimal growth phenotypes (Cheong et al, 2017). The finer aspects of the interactome circuitry of CYP20-3 with reductants (TRXs, 2CPs, and SAT1) will further delineate the regulatory dynamics of balancing acts in optimizing plant fitness, under various forms of environmental pressures.

## Materials and Methods

### Preparation of recombinant proteins

Coding sequences for the mature protein region of 2CPA (At3g11630), 2CPB (At5g06290), NTRC (At2g41680), and SRX (At1g31170) were cloned into the pET28a vector (Novagen) using *BamHI*/*HindIII* (tagged version)

and/or *NdeI*/*HindIII* (tag-free version). Point mutations of 2CPB were introduced using the QuikChange II site-directed mutagenesis kit (Agilent) according to the manufacturer's instructions. The proteins were then expressed in *E. coli* BL21 (DE3) and purified by a nickel-column (Ni-NTA; QIAGEN) as previously described (Cheong et al, 2017). To remove the His-tag, purified 2CPs and mutant 2CPs were incubated with thrombin protease. The resulting proteins contain additional four nonnative residues (Gly, Ser, His, and Met) at the N terminus of the protein. Primers used for plasmid constructions and site mutagenesis are listed in Table S1.

### S-glutathionylation of 2CPs

Typically, S-glutathionylation reactions were conducted by incubating 1 or 2 $\mu M$ 2CPs with 1.0 mM GSH, GSSG, or GSNO in 50 mM Tris–HCl buffer (pH 7.5) at 25°C for 30 min, although some reactions varied GSH concentrations (0.5–10 mM), incubation times (0.5–30 min), or buffer pH (7.0–8.0).

### Preparation of GSH-glutathionylated and oxidized 2CPs

In 50 mM Tris–HCl (pH 7.5) buffer, 10 $\mu M$ 2CPs was S-glutathionylated for 30 min by 10 mM GSH, or oxidized for 15 min by 0.1 mM $H_2O_2$. Following the reactions, excess GSH and $H_2O_2$ were removed using size-exclusion chromatography (Sephadex G-25 medium; GE Healthcare) and stored at 4°C until use.

## Peroxidase activity assay

Reduction of $H_2O_2$ by proteins was quantified via the eFOX assay method (Cheeseman, 2006). Briefly, the assay was performed at 37°C in 50 mM Tris buffer (pH 7.5) containing 50 mM NaCl with 5 $\mu M$ 2CPs. Each reaction was initiated by the addition of 50 $\mu M$ $H_2O_2$, then incubated for 10 min, and terminated by 2% (vol/vol) trichloroacetic acid. A volume of 500 $\mu l$ eFOX reagent (250 $\mu M$ $Fe(NH_4)_2(SO_4)_2$, 100 $\mu M$ sorbitol, 100 $\mu M$ xylenol orange, and 1% [vol/vol] in 20 mM $H_2SO_4$) was then mixed with 100 $\mu l$ of the reaction solution, and the reduction in $H_2O_2$ levels was tracked spectrophotometrically by measuring the difference in absorbance between 550 and 800 nm.

## Chaperone activity assay

The chaperone activity of 2CPs was measured using citrate synthase as a substrate (Bhuwan et al, 2017). Briefly, 10 $\mu M$ 2CPs was incubated at 45°C in 50 mM potassium phosphate buffer (pH 7.2). After temperature stabilization for 15 min, citrate synthase (1 $\mu M$; Sigma-Aldrich) was added, and the increase in absorption at 360 nm was monitored with a spectrophotometer.

## Plant materials

*Arabidopsis thaliana* WT plants (Col-0) and homozygous T-DNA insertion mutants in *2CPA* (SALK_065264; Kangasjärvi et al, 2008 [*2cpaI*]; Ishiga et al, 2012 [*2cpaII*]), *2CPB* (SALK_017213; Ishiga et al, 2012), *2CPA/2CPB* (Pulido et al, 2010), *NTRC* (SALK_012208; Stenbaek et al, 2008), *CYP20-3* (SALK_120440; Dominguez-Soils et al, 2008), and *phytoalexin deficient 2* (*pad2*; Glazebrook & Ausubel, 1994) were used in this study. Plants were grown in a chamber with a 12-h day cycle (80–100 $\mu E/m^2/s$) at 22°C and 60–80% relative humidity. For HS experiments, 2 h after dawn, plants were transferred to a growth chamber with these same light conditions but with the temperature set at 42°C.

## Total protein extraction

Leaf tissues of Arabidopsis were immersed in liquid $N_2$ and ground to a powder using a mortar and pestle. Ground tissue was dissolved into two volumes of 50 mM potassium phosphate buffer (pH 7.2) containing protease inhibitor cocktails (Sigma-Aldrich), agitated for 60 min, and centrifuged for 30 min at 10,000*g*. The supernatant was collected and immediately used for Western blot analyses. Note that all extraction steps were carried out at 4°C.

## In vitro and ex vivo Western blot analysis

To assess the quaternary structure of 2CPs, total protein extracts freshly prepared or recombinant 2CPs[GS] were resolved by SDS/PAGE and electroblotted onto polyvinylidene fluoride membranes (Millipore). The resulting blots were probed with protein G–purified polyclonal rabbit anti-2CPA antibody (1:7,500; MyBioSource) for 2 h, or monoclonal mouse anti-GSH antibody (1:3,000; Enzo Life Science) for 16 h (for in situ WB) or 2 h (for in vitro WB), and visualized by chemiluminescence (ECL kit; GE Healthcare). If needed, Ponceau-S red staining was used to verify equal loading.

## In situ $H_2O_2$ measurement

Measurements of endogenous concentrations of $H_2O_2$ and peroxidase activities in Arabidopsis leaf tissues were measured by using the Red Hydrogen Peroxide Assay Kit (Enzo Life Science) according to the manufacturer's instructions. The harvested samples (100 mg) were ground in liquid $N_2$ and suspended in 200 $\mu l$ of 20 mM sodium phosphate buffer (pH 7.4). The mixture was centrifuged at 9,500*g* for 10 min at 4°C, and the supernatant was used for the subsequent assays.

## Quantitative RT-PCR

Total leaf RNA was prepared using TRIzol reagent (Invitrogen) and RNase-free DNase (RQ1; Promega) according to the manufacturer's instructions. RNA qualities were assessed by agarose gel electrophoresis and NanoDrop ($A_{260}/A_{280}$ > 1.8 and $A_{260}/A_{230}$ > 2.0; Udvardi et al, 2008). cDNA synthesis was performed by using an oligo(dT) reverse primer and a reverse transcriptase (qScript; QuantaBio). Quantitative PCR was performed with the PerfeCT SYBR Green FastMix Reaction Mixes (QuantaBio) in the CFX96 Touch (Bio-Rad) PCR system cycled 40 times by using gene-specific primer sets (Table S1). The annealing temperatures for the primer pairs were 53°C. Data were quality-controlled; normalized against three reference genes, *polyubiqutin* (*UBC*), *GAPDH*, and *protein phosphatase 2A* (*PP2A*) (Czechowski et al, 2005; Sanchez-Villarreal et al, 2013); and statistically evaluated using qbasePLUS 3.0 (Ramakers et al, 2003). Primers used for qRT-PCR are listed in Table S1.

# Supplementary Information

# Acknowledgements

We thank KJ Dietz for sharing pCRT7/NT-TOPO encoding 2CPA (At3g11630) and FJ Cejudo and K Mysore for providing transfer-DNA insertion lines: *2cpaI*, *2cpaII*, *2cpb*, *2cpa/2cpb*, and *ntrc*. This work was supported by the Alabama Agricultural Experiment Station (Auburn University), the Hatch Program of the National Institute of Food and Agriculture (United States Department of Agriculture), the office of the Vice President for Research and Economic Development (Auburn University), the Alabama Cotton Commission, and the Alabama Farmers Federation.

## Author Contributions

W Liu: conceptualization, data curation, formal analysis, validation, investigation, visualization, and writing—original draft.
I Barbosa Dos Santos: conceptualization, data curation, validation, investigation, visualization, and writing—original draft.
A Moye: investigation.
S-W Park: conceptualization, data curation, formal analysis, supervision, funding acquisition, investigation, and writing—original draft, review, and editing.

## Conflict of Interest Statement

The authors declare that they have no conflict of interest.

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
