## [Reviewer comments · Life Science Alliance]

Life Science Alliance

CYP20-3 Deglutathionylates 2-CysPRX A and Suppresses Peroxide Detoxification during Heat Stress

Wenshan Liu, Izailda Barbosa Dos Santos, Anna Moye, and Sang-Wook Park

DOI: <https://doi.org/10.26508/lsa.202000775>

Corresponding author(s): Sang-Wook Park, Auburn Univeristy

Review Timeline:	Submission Date:	2020-05-12
	Editorial Decision:	2020-05-13
	Revision Received:	2020-07-06
	Editorial Decision:	2020-07-15
	Revision Received:	2020-07-17
	Accepted:	2020-07-21

Transaction Report:

Please note that the manuscript was previously reviewed at another journal and the reports were taken into account in the decision-making process at Life Science Alliance.

Referee #1 Review

Report for Author:

The manuscript „CYP20-3 Deglutathionylates 2-CysPRX A and Suppresses Peroxide Detoxification during Heat Stress" by Liu et al. describes an activation mechanism of 2CPA and B by GSH glutathionylation and a deactivation mechanism mediated by CYP20-3. The authors carefully studied the differences between 2CPA and B and pinpoint two amino acids (out of seven, which are different between both proteins). They show the modification by GSH in vitro and in vivo and carry out careful studies of the dynamics of 2CP dimerization and multimerization and the effect of CYP20-3 - mostly in vitro.

I am genuinely impressed by the amount and quality of data. However, some aspects of the manuscript are not well supported by in planta experiments. While the in vitro molecular analysis allows some exciting mechanistic insights in the regulation of 2CPs, in vivo validation would be

required to appreciate the "real world" relevance. The occasionally somewhat confusing organization of the manuscript (it is simply difficult to read) together with the description of the experiments contributes to this impression.

That being said, if the authors can make their manuscript more accessible and also provide in vivo support for their mechanistic in vitro work, this manuscript would make an outstanding contribution on how 2CPs and GSH contribute to signalling in plants!!

Specific comments:

1. Abstract and introduction are very specific and technical and require the reader to be familiar with 2CPs and their regulation in order to understand the background to the work. It should not be hard for the authors to make the introductory section more accessible to the reader which would also significantly raise the impact of their work.

2. Another somewhat confusing aspect of the introduction is that Figure S1, S2 and S3 are referenced in the introduction - i.e. as background information. I got the impression that this was novel data and should thus be described in the results section and the introduction. If this is validation of previously published data this should be highlighted better.

In the results section I found it hard to immediately understand for all experiments whether they represent an in vivo approach to elucidate the regulation of 2CPs or an in vitro reconstitution of regulatory mechanisms. It can be figured out, however, it would be preferable if this would be immediately understandable for the reader.

3. "This mutant 2CPBV106W/109V indeed behaved like 2CPA" (page 6 of the manuscript): This is an exciting finding. How about the corresponding point mutations in 2CPA? Could that convert it into a "2CPB-like behaviour"? Along the same lines: can this be replicated in vivo by transforming the point mutation constructs into 2cpa or 2cpb mutant plants? This should be doable since the authors have mutant lines for 2cpa and 2cpb as well as an antibody recognizing both proteins (Figure S2A and B). Would the point mutation constructs in the difference 2cp mutant plant lines complement their phenotypes (for example hydrogen peroxide production in 2cpa)?

4. Figure 1, panels C and D: I assume "GST" should be GSH?

5. "The latter further supports a unique and autonomous activity of GSH in activating redox signaling, independently from GSSG-mediated Sglutathionylation and its oxidative signaling (Xiong et al, 2011; Grek et al, 2013), that relays metabolic signaling during the stress-responsive activation of defense and acclimation pathways." (page 7 of the manuscript) - this is only based on the (in vitro) 2CP data and should either be rephrased to reflect that or supported by in vivo evidence.

6. "Herein, TRX and NTRC exhibited minimal reductase activity, exhibiting little if any effect on the quaternary structure of 2CPsGS." (page 8 of the manuscript) - Figure 3 panel D suggests that addition of NTRC leads to more 2CPB icosamer. Is that correct? If yes this could be rephrased, if not perhaps a clearer image with better resolution could be supplied.

7. "Though, it was noticeable that the supplement of SRX causes a slight delay in the gel mobility of partial 2CPBsGS (Fig 3F), indicating that SRX may be able to target the S-S bond forming 2CPBGS icosamers (Fig 1D)." (page 8 of the manuscript): Would 2CPB cysteine mutants (C -> S) be helpful to visualize this better? In panel 3F I can only observe the decamers but not really the icosamers. Generally, the icosamer is nicely visible in Figure 1 but very hard to see in the other figures.

8. "Indeed, the deglutathionylation of 2CPAGS paralleled an attenuation of its (peroxidase activity, reducing H₂O₂ (Fig 5A)" (page 10 of the manuscript): The experiment has been carried out under "heat stress" but no control under ambient temperatures is shown.

9. "The enhanced redox capacity in turn coordinates the expression of a subset of OPDA-responsive genes (ORGs), including HSP17.6, as well as CYP81D11 (Fig 5D and Appendix Fig S9)." (page 10 of the manuscript): The results in Figure 5D suggest that HSP17.6 is only partially dependent on OPDA (or the action of CYP20-3). I suggest to use more than two marker genes for the action of OPDA.

10. "These results concur with the conclusion that CYP20-3-dependent OPDA (defense) signaling counteracts the photosynthetic (growth) mechanisms, drawing a new insight into the mode of "growth and defense trade-offs" that optimizes energy productions and distributions during defense mechanisms (e.g., HS responses), maximizing plant growth and survival processes." (page 10 of the manuscript): I don't precisely follow the authors argumentation here. There is little data on the connection to OPDA signalling in the manuscript at present. This needs to be much better explained to the reader and possibly supported by experimental evidence.

11. Technical note: qRT-PCR experiments are normalized to a single reference gene. An experimental design using at least three reference genes has been recommended for past years and for a high-quality publication should be standard procedure.

Referee #2 Review

Report for Author:

First of all, I would like to state that I was not aware at the time of reading the manuscript that this had already been rejected and my reviewing it was within the procedure of rebuttal/re-evaluation. Thus, I was not influenced by either the first editorial decision or the rebuttal by the authors.

Liu et al describe a mechanism by which a chloroplastic cyclophilin is involved in the growth/defense response of plants under heat stress. The authors state that OPDA as a stress-response molecule binds to CYP20-3 and thereby leads to decrease of 2CPA peroxidase activity and at the same time the increase of SAT1 action to build up thiol synthesis.

General remark regarding the quality of the manuscript:

The text is often hard to read and the use of words often in the wrong context, e.g. "metabolize" is not a process that changes protein structures but usually refers to catabolizing something such as glucose to ATP in glycolysis. Another example is "entertaining" plants at 12h light/12h dark - I very much doubt that growing in climatic chambers is entertaining to the plants. There are more cases like this, which confuse the reader.

Introduction:

The introduction is well linked to the rest of the manuscript, but I miss several facts that should be mentioned here. First the connection of 2CP and NTRC (especially since there are results later regarding their interaction), second the confirmed and important role of OPDA in hormone signaling. Researchers from that field will wonder upon reading the manuscript why it is not mentioned anywhere that OPDA is the precursor of jasmonic acid and thus heavily involved in stress response to cold, wounding and biotic stresses. It needs to be discussed how OPDA is sequestered to the different biochemical processes - binding to CYP20-3 would surely prevent export and conversion

to jasmonate.

Results:

Figure 3/RNA expression: the authors argue their case with co-expression of RNA and deduce protein activity from that. This is simple not feasible. It has been shown many times now (among others by the Dietz lab, from whom they even got material) that RNA amount has in ~50% nothing to do with protein amount, let alone protein activity. Co-expression data can be valuable nonetheless, but this needs to be discussed and the phrasing adapted. The same holds true for Figure 5.

Discussion: please refer to my comments above!

Regarding the novelty of the presented data, details about 2CP function in relation to OPDA-binding to CYP20-3 are indeed new as is the differentiation of 2CPA and 2CPB. This manuscript points to a yet unrecognized role of these components in the growth/defense regulation of plant metabolism.

However, there are points arguing against publication in this journal - the methods used here are quite basic, repetitive and not in any way sophisticated. The results will only be interesting to a specialized reader community from the plant field.

May 13, 2020

Re: Life Science Alliance manuscript #LSA-2020-00775-T

Dr. Sang-Wook Park
Auburn Univeristy
Entomology and Plant Pathology
209 Rouse Life Science Bldg.
Auburn, Alabama 36849

Dear Dr. Park,

Thank you for transferring your manuscript entitled "CYP20-3 Deglutathionylates 2-CysPRX A and Suppresses Peroxide Detoxification during Heat Stress" to Life Science Alliance. The manuscript was assessed by expert reviewers at another journal, and the editors transferred those reports to us with your permission.

The reviewers who evaluated your study elsewhere appreciated your findings, but would have expected a further reaching demonstration of their (potential) *in vivo* relevance. This concern does not need to get fully addressed for publication in Life Science Alliance, and we would thus like to invite you to submit a revised version of your manuscript to us, based on the reviewer reports already at hand. Please provide a point-by-point response and accordingly drastic changes to the manuscript text (both reviewers). The requests for controls and technical comments should get addressed (rev#1), too.

Thank you for this interesting contribution to Life Science Alliance. We are looking forward to receiving your revised manuscript.

Sincerely,

B. MANUSCRIPT ORGANIZATION AND FORMATTING:

Dear Dr. Leibfried,

We are very pleased to submit our revised manuscript (#LSA-2020-00775-T, Liu et al.) for consideration of publication in your esteemed journal, *Life Science Alliance*. We wish to sincerely thank the referees for their very constructive comments and concerns. All these have been addressed in the revised manuscript. In addition, the text of the manuscript has been significantly revised for clarity and proficiency pursuant to your editing. Please find our responses to the referees below.

Response to Referee #1

We have addressed the referee's comments and concerns regarding the sufficient introduction and discussion. Below we have described all the changes and modifications made in the revised manuscript as per the suggestions. A = our answer/response to the reviewer's suggestions/comments which are lettered in lowercase (e.g. a, b, c, etc).

- a. Abstract and introduction are very specific and technical and require the reader to be familiar with 2CPs and their regulation in order to understand the background to the work. It should not be hard for the authors to make the introductory section more accessible to the reader which would also significantly raise the impact of their work.
- A, We agree with the referee's suggestion, and have revised and substantially rewritten both abstract and introduction to be more accessible for readers. These changes/insertions include;
 - 1) "The latter is a thiol-based peroxidase, protecting and optimizing photosynthesis by reducing its toxic byproducts (e.g., H₂O₂)", (page #2 line #8 in the revised manuscript).
 - 2) "The peroxidase cycle starts by their redox-active cysteine (Cys; the peroxidatic Cys, C_P) in the catalytic center is oxidized to sulfenic acid (C_P-SOH) by a broad set of peroxide substrates. The C_P-SOH residue then reacts either with another Cys of the same or an adjacent subunit, or with another thiol-containing compound, to form an inter- or intramolecular disulfide (C_P-S-S-C_R), which is later reduced (via a mixed disulfide with a reductant) to reform the thiolate (Perkins et al, 2015; Liebthal et al, 2019)", (page #2 line #18 in the revised manuscript).

3) “However, 2CPs are known to typically form an obligatory homodimer as the C_P from one monomer is connected via a disulfide bond to the resolving Cys (C_R) located at another monomer. The oxidation of Cys^P then, in consequence, deactivates the catalytic activity of 2CPs. Thus, 2CP dimers require electron donors such as an NADPH-dependent thioredoxin reductase C (NTRC), thioredoxins (TRXs) and/or cyclophilin 20-3 (CYP20-3), which reduces (activates) them to be able to metabolize the detoxification of a toxic byproduct in photosynthesis (i.e., H₂O₂), and the activation of Calvin cycle enzymes such as a fructose 1,6-bisphosphatase (Dietz et al, 2006; Laxa et al, 2007; Caporaletti et al, 2007; Muthuralingam et al, 2009; Liebthal et al, 2016).” (page #3 line #4 in the revised manuscript).

4) “On the other hand, 2CPs could display an array of oligomeric structures upon cellular positions and conditions, as well as other post-translational modifications (PTMs; Perkins et al, 2015; Liebthal et al, 2019). Under oxidative stress, 2CPs could be overoxidized and form a homodecameric complex that disables their peroxidase activity, but instead gains a chaperon activity to protect cellular molecules against oxidative damage (Liebthal et al, 2019). However, recent studies with human PrxI (huPrxI) and pea 2CP have argued that 2CPs are rather S-glutathionylated by GSSG during oxidative stress, protecting their quaternary structures to remain as dimers and, as a result, enzymatically inactivated status (Park et al, 2009, 2011; Calderón et al, 2017).” (page #3 line #16 in the revised manuscript).

5) “OPDA is a primary precursor of (-)-jasmonic acid (JA), able to trigger an autonomous signaling pathway that regulates unique subsets of jasmonate-responsive genes, activating and fine-tuning plant defense responses, as well as growth processes (Böttcher & Pllmann, 2009; Dave & Graham, 2012). Its distinctive activity was first described by the pathoanalyses of a mutant *Arabidopsis* (*opr3*) arresting the conversion of OPDA to JA (Stintzi et al, 2001). Wild type (WT)-like resistance of *opr3*, in contrast to decreased resistance in mutant plants disrupting trienoic-fatty acid biosynthesis (*fad3/7/8*) and the octadecanoid pathway (*dde2* and *aos*), against fungal and insect infections underlined the essential roles of OPDA signaling in plant defense responses in the absence of JA and JA-Ile (Stintzi et al, 2001; Zhang and Turner, 2008; Stotz et al, 2011). Following studies with several mutant plants suppressing or impairing JA production (e.g. *siOPR3*, *OPR3-RNAi*, *cts-2/opr3* and *acx1*) or OPDA signaling (*cyp 20-3*) further substantiated that OPDA signaling is crucial in basal defense responses against a variety of pathogenic fungi and insects such as *Alternaria brassicicola*, *Botrytis cinerea*, *Scierotinia sclerotiarum*, *Nilaparvata lugens*, *Manduca sexta* and *Bradysia impatiens*, as well as seed germination, embryogenesis and balancing abscisic acid signaling (Dave et al, 2011; Goetz et al, 2012; Park et al, 2013; Guo et al, 2014; Bosch et al, 2014; Scalschi et al, 2015)” (page #4 line #6 in the revised manuscript).

- b. Another somewhat consuming aspect of the introduction is that Figure S1, S2 and S3 are referenced in the introduction – i.e. as background information. I got the impression that this was novel data and should thus be described in the results section and the introduction. If this is validation of previously published data this should be highlighted better.

In the result section I found it hard to immediately understand for all experiments whether they represent an in vivo approach to elucidate the regulation of 2CPs or an in vitro reconstitution of regulatory mechanisms. It can be figured out, however, it would be preferable if this would be immediately understandable for the reader.

- A, We agree with the referee’s suggestion, and have incorporated the results from the Figure S1, S2 and S3 into appropriate areas in the revised result sections, as it reads;

1) “Recently, emerging evidences have elucidated a critical role of redox-mediated PTM in resolving the cellular property and modus operandi of 2CPs (Park et al, 2009, 2011; Calderón et al, 2017). In

agreement, our preparatory analyses exhibited that 2CPs can uniquely bind a negatively charged tripeptide GSH, a major nonprotein thiol-antioxidant in plants (**Fig S1A**). GSH-binding (hereafter, called GSH-glutathionylation) then differentially modulates the conformational states of 2CPs (Figs 1A lane 1, 2, 5 and 6, and **S2**), stimulating the monomerization and peroxidase activity of 2CPA, while decamerizing and accentuating the chaperone activity of 2CPB (**Fig S3**, Lee et al, 2015)", (page #5 line #22 in the revised manuscript).

2) "Next, we investigated whether cellular redox homeostasis may influence the rate GSH-glutathionylation of 2CPs. --- In a resting state, the chloroplasts that produce SA and OPDA maintained GSH:GSSG ratios of 14:1 (Koffler et al, 2013), and these escalated up to $\geq 28:1$ under stress conditions (Park et al, 2013). The enhanced redox capacity then stimulates the GSH-glutathionylation and monomerization of 2CPA^{GS} (Fig 2A lane 3-5), which in turn likely accentuated its peroxidase activity (**Fig S3B**)" (page #7 line #19 in the revised manuscript).

3) "This explains the HS-induced accumulation of, already abundant, 2CPB^{GS} (Fig 5C) that constitutes a stable, decameric conformation (Fig 1) conferring chaperon activity (**Fig S3C**)" (page #10 line #16 in the revised manuscript).

- c.** "This mutant 2CPBV106I/I109V indeed behaved like 2CPA" (page 6 of manuscript): This is an exciting finding. How about the corresponding point mutations in 2CPA? Could that convert it into a "2CPB-like behaviours"? Along the same lines: can this be replicated in vivo by transforming the point mutation constructs into *2cpa* and *2cpb* as well as an antibody recognizing both proteins (Figure S2A and B). Would the point mutation constructs in the difference *2cp* mutant plant lines complement their phenotypes (for example hydrogen peroxide production in *2cpa*)?
- A,** As per the referee's suggestion, we have examined and presented the behavior of a WT and mutant (*I106V/V109I*) 2CPAs upon GSH-glutathionylation (Fig 1B lanes 13 and 14 in the revised manuscript), exhibiting 2CPB-like pattern.

We also agree that the complementation study is a logical next step to further validate the physiological role of 2CPA. However, because of current circumstantial and time constraints, we have discussed with an editor, and instead substantiated the role of 2CPA by using the second mutant allele (*2cpaII*; MPML 25:294) as well as a double KO mutant (*2cpa/2cpb*; J Ext Bot 61:4043). These results clearly underpinned the important role of 2CPA in the removal of H₂O₂ during heat stress responses (Fig 5B in the revised manuscript). We hope this is acceptable to the reviewer.

- d.** Figure 1, panels C and D: I assume "GST" should be GSH?.
- A,** The type-error has been corrected in the revised Figure 1.
- e.** "The latter further supports a unique and autonomous activity of GSH in activating redox signaling, independently from GSSG-mediated S-glutathionylation and its oxidative signaling (Xiong et al. 2011; Grek et al, 2013), that relays metabolic signaling during the stress-responsive activation of defense and acclimation pathways." (page 7 of the manuscript) - this is only based on the (in vitro) 2CP data and should either be rephrased to reflect that or supported by in vivo evidence.
- A.** As per the referee's suggestion, we have presented new ex vivo experiment, demonstrating that CYP20-3-dependent OPDA signaling and subsequent increases in GSH:GSSG ratio (PNAS 11:9559) play a critical role in regulating the conformational state (rates) of 2CP^{GS} to accentuate its peroxidase

activity during wound defense and acclimation processes (Fig S5 in the revised manuscript). We have also rephrased the last sentence, as it reads in the revised manuscript (page #8, line #8) “The latter further supports a unique and autonomous activity of GSH in activating redox signaling, independently from GSSG-mediated S-glutathionylation and its oxidative signaling (Xiong et al, 2011; Grek et al, 2013), that relays hormone (e.g., OPDA; **Fig S5**) signaling_during the stress-responsive activation of defense and acclimation pathways.”

- f. “Herein, TRX and NTRC exhibited minimal reductase activity, exhibiting little if any effect on the quaternary structure of 2CPs^{GS}.” (page 8 of the manuscript) – Figure 3 panel D suggests that addition of NTRC leads to more 2CPB icosamer. Is that correct? If yes this could be rephrased, if not perhaps a clearer image with better resolution could be supplied.
- A. We appreciate referee’s suggestion. We have repeated the assay, and also reassessed earlier results. To the end, we have carefully concluded that NTRC showed little effect on the icosameric structure of 2CPB^{GS}, and thus – as per the referee’s suggestion – replaced the image with one of newer data with an improved resolution (Fig 3D in the revised manuscript).
- g. “Though, it was noticeable that the supplement of SRX causes a slight delay in the gel mobility of partial 2CPBs^{GS} (Fig 3F), indicating that SRX may be able to target the S-S bond forming 2CPB^{GS} icosamers (Fig 1D).” (page 8 of the manuscript): Would 2CP cysteine mutants (C→S) be helpful to visualize this better? In panel 3F, I can only observe the decamers but not really the icosamers. Generally, the icosamer is nicely visible in Figure 1 but very hard to see in the other figures.
- A, We appreciate referee’s insights. We - first of all - have replaced the Fig 3F with one with an improved resolution.

On the other hand, we re-examined and found that a mutant 2CPB does not improve the visibility of 2CPB^{GS} icosamers. As a referee suggested, we have validated that 2CPB cysteine mutant (converting C¹⁷⁵ to S) is able to form icosamers upon the GSH-glutathionylation (Fig 1A in this letter). However, the level icosamerization of 2CPB•C175S^{GS} was similar to that of WT 2CPB^{GS} (Fig 1B in this letter). Moreover, we have detected that the initial amounts of icosameric WT and mutant 2CPBs^{GS} (these are shown in Fig 1 in the revised manuscript) were noticeably reduced by the dialysis process for the following experiments (Fig 1B in this letter; and these are shown in Figs 3 and 4 in the revised manuscript). Taken together, we conclude that 2CPB•C175S^{GS} does not significantly improve the visibility of 2CPB^{GS} icosamers, and believe that further studies are needed to comprehend the biochemical characteristics and stability of 2CPB^{GS} icosamers.

[Figure removed by Editorial Staff per authors' request]

- h. “Indeed, the deglutathionylation of 2CPA^{GS} paralleled an attenuation of its peroxidase activity, reducing H₂O₂ (Fig 5A)” (page 10 of the manuscript): The experiment has been carried out under “heat stress” but no control under ambient temperatures is shown.

- A, We have included an control experiment; the peroxidase activity of 2CPA^{GS} upon the incubation of CYP20-3 at ambient temperatures (22 °C, Fig 5A in the revised manuscript). CYP20-3 exhibited little, if any, effect on the peroxidase activity of 2CPA^{GS}.
- i. The enhanced redox capacity in turn coordinates the expression of a subset of OPDA-responsive genes (ORGs), including *HSP17.6*, as well as *CYP81D11* (Fig 5D and Appendix Fig S9).” (page 10 of the manuscript): The results in Figure 5D suggest that HSP17.6 is only partially dependent on OPDA (or the action of CYP20-3). I suggest to use more than two marker genes for the action of OPDA.
- A, As per the referee’s suggestion, we have examined and presented the attenuation of another OPDA marker gene (*HSP70*, Plant Cell 20:768) in *cyp20-3* KO mutant during heat stress (**Fig S10B** in the revised manuscript).
- j. “The results concur with the conclusion that CYP20-3-dependent OPDA (defense) signaling counteracts the photosynthetic (growth) mechanisms, drawing a new insight into the mode of “growth and defense trade-offs” that optimizes energy productions and distributions during defense mechanisms (e.g., HS responses), maximizing plant growth and survival processes.” (page 10 of the manuscript): I don’t precisely follow the authors argumentation here. There is little data on the connection to OPDA signaling in the manuscript at present. This needs to be much better explained to the reader and possibly supported by experimental evidence.
- A. We agree with the referee’s comment, and removed the previous discussion on the role of OPDA signaling. The revised manuscript now reads (page #10 line #24) “These results support the versatile activity of CYP20-3 in OPDA signaling, which conveys the activation of disease resistance against *Alternaria brassicicola*, and defense responses to different abiotic stresses such as wounding and HS (Park et al, 2013; Figs 5D and Fig S10)”.
- k. Technical note: qRT-PCR experiments are normalized to a single reference gene An experimental design using at least three reference genes has been recommended for past years and for a high-quality publication should be standard procedure.
- A. As per the referee’s suggestion, we have examined and presented the qRT-PCR experiments (revised Figs 3G, 3H, 5C, 5D, and S10; revised Materials and Methods, page #17 line #11), normalized against three reference genes, *polyubiquitin (UBC)*, *glycer-aldehyde 3-phosphate dehydrogenase (GAPDH)* and *protein phosphatase 2A (PP2A)*, and evaluated using qbasePLUS 3.0 (Ramakers et al, 2003).

Response to referee #2

- a. The text is often hard to read and the use of words often in the wrong context, e.g. “metabolize” is not a process that changes protein structures but usually refers to catabolizing something such as glucose to ATP in glycolysis. Another example is “entertaining” plants at 12h light/12 dark – I very much doubt that growing in climatic chambers is entertaining to the plants. There are more cases like this, which confuse the reader.
- A, As per referee’s suggestion, we have replaced “metabolize” with “convert” (page #8, line #14 in the revised manuscript), and “entertained” with “grown” (page #9 line #1 in the revised manuscript), and carefully revised the manuscript.

Introduction:

- b. The introduction is well linked to the rest of the manuscript, but I miss several facts that should be mentioned here. First the connection of 2CP and NTRC (especially since there are results later regarding their interaction), second the confirmed and important role of OPDA in hormone signaling. Researchers from that field will wonder upon reading the manuscript why it is not mentioned anywhere that OPDA is the precursor of jasmonic acid and thus heavily involved in stress response to cold, wounding and biotic stresses. It needs to be discussed how OPDA is sequestered to the different biochemical processes – binding to CYP20-3 would surely prevent export and conversion to jasmonate.
- A. As per the referee's suggestions, we have introduced (1) the connection of 2CP and NTRC and (2) the role of OPDA in hormone signaling in the revised introduction. In addition, we have discussed (3) the impact of OPDA binding to CYP20-3 in the revised discussion.

1) The revised manuscript reads (page #3, line #7) "Thus, 2CP dimers require electron donors such as an NADPH-dependent thioredoxin reductase C (NTRC), thioredoxins (TRXs) and/or cyclophilin 20-3 (CYP20-3), which reduces (activates) them to be able to metabolize the detoxification of a toxic byproduct in photosynthesis (i.e., H₂O₂), and the activation of Calvin cycle enzymes such as a fructose 1,6-bisphosphatase (Dietz et al, 2006; Laxa et al, 2007; Caporaletti et al, 2007; Muthuramalingam et al, 2009; Liebthal et al, 2016)."

2) In page #4, line #6: "OPDA is a primary precursor of (-)-jasmonic acid (JA), able to trigger an autonomous signaling pathway that regulates unique subsets of jasmonate-responsive genes, activating and fine-tuning plant defense responses, as well as growth processes (Böttcher & Pollmann, 2009; Dave & Graham, 2012). Its distinctive activity was first described by the pathoanalyses of a mutant *Arabidopsis* (*opr3*) arresting the conversion of OPDA to JA (Stintzi et al, 2001). Wild type (WT)-like resistance of *opr3*, in contrast to decreased resistance in mutant plants disrupting trienoic-fatty acid biosynthesis (*fad3/7/8*) and the octadecanoid pathway (*dde2* and *aos*), against fungal and insect infections underlined the essential roles of OPDA signaling in plant defense responses in the absence of JA and JA-Ile (Stintzi et al, 2001; Zhang and Turner, 2008; Stotz et al, 2011). Following studies with several mutant plants suppressing or impairing JA production (e.g. *siOPR3*, *OPR3-RNAi*, *cts-2/opr3* and *acx1*) or OPDA signaling (*cyp 20-3*) further substantiated that OPDA signaling is crucial in basal defense responses against a variety of pathogenic fungi and insects such as *Alternaria brassicicola*, *Botrytis cinerea*, *Scierotinia sclerotiarum*, *Nilaparvata lugens*, *Manduca sexta* and *Bradysia impatiens*, as well as seed germination, embryogenesis and balancing abscisic acid signaling (Dave et al, 2011; Goetz et al, 2012; Park et al, 2013; Guo et al, 2014; Bosch et al, 2014; Scalschi et al, 2015).

Under stressed conditions, OPDA - accumulates in the chloroplasts - binds and promotes CYP20-3 to transfer electrons from the photosystem I (PSI) via TRXs (type-f2 and -x) towards 2CPs (Motohashi et al, 2001; Laxa et al, 2007; Dominguez-Soils et al, 2008; Cheong et al, 2017) or a serine acetyltransferase 1 (SAT1, Dominguez-Soils et al, 2008; Park et al, 2013). Reduction of 2CPs then controls peroxide (photooxidant) detoxifications and photosynthetic carbon metabolisms (Dietz et al, 2006; Caporaletti et al, 2007), whereas the activation of SAT1 stimulates the plastid sulfur assimilation which leads to the production of Cys and thiol metabolites (e.g., glutathione; GSH), and the buildup of cellular reduction potential (Park et al, 2013). The enhanced reduction capacity in turn coordinates the expression of a subset of OPDA-responsive genes (ORGs) and general defense regulators (e.g., *glutaredoxin 480*) in controlling basal and race-specific (local and systemic) resistances and defense responses against various abiotic stresses (Mou et al, 2003; Park et al, 2013)."

3) In page #13, line #5: "Note that our jasmonate quantifications in *cyp20-3* KO mutants (Park et al, 2013) suggest that, in a resting states, CYP20-3 could sequester OPDA and reduce downstream jasmonate productions, but the increased accumulations of OPDA under stress conditions could circumvent the effect of its binding to CYP20-3, exhibiting little difference in JA accumulations between WT and *cyp20-3*, together proposing that OPDA and JA signaling are activated in parallel and/or accumulatively in defense responses."

Results:

- c. Figure 3/RNA expression: the authors argue their case with co-expression of RNA and deduce protein activity from that. This is simple not feasible. It has been shown many times now (among others by the Dietz lab, from whom they even got material) that RNA amount has in ~50% nothing to do with protein amount, let alone protein activity. Co-expression data can be valuable nonetheless, but this needs to be discussed and the phrasing adapted. The same holds true for Figure 5.
- A. We agree with the referee's suggestion and have incorporated additional descriptions. The revised manuscript reads (page #9, line #4) "Caveat is that 2CPs are highly abundant (~0.6% of the total plastid proteins), and determined to exhibit slow-turnover rates (Horling et al, 2003; Dietz et al, 2006). Hence, the co-expression of 2CPs and SRX may not, at once, tie in their physiological and functional interactions."
In addition, (page #10, line #16) "This explains the HS-induced accumulation of, already abundant, 2CPB^{GS} (Fig 5C) that constitutes a stable, decameric conformation (Fig 1) conferring chaperon activity (Fig S3C)."

Having addressed all of the concerns of the reviewers, we hope that the revised manuscript is now acceptable for publication in *Life Science Alliance*.

July 15, 2020

RE: Life Science Alliance Manuscript #LSA-2020-00775-TR

Dr. Sang-Wook Park
Auburn Univeristy
Entomology and Plant Pathology
209 Rouse Life Science Bldg.
Auburn, Alabama 36849

Dear Dr. Park,

Thank you for submitting your revised manuscript entitled "CYP20-3 Deglutathionylates 2-CysPRX A and Suppresses Peroxide Detoxification during Heat Stress". Your manuscript was re-reviewed by one of the original referees at another journal, and their report is attached below. We would be happy to publish your paper in Life Science Alliance pending final revisions necessary to meet our formatting guidelines.

- please address the remaining reviewer's comments and indicate in the results section in the text and in the figure legends when an experiment has been carried out with bacterially produced protein
- please add ORCID ID for corresponding author-you should have received instructions on how to do so
- please upload your supplementary figures as singular files and add the supplementary figure legends to the main manuscript text
- please provide your tables as a separate file in editable docx or excel format
- please check your Figure Callouts - you have a callout for Fig. S1A, but there is not a panel A in figure S1

A. FINAL FILES:

B. MANUSCRIPT ORGANIZATION AND FORMATTING:

Sincerely,

Reilly Lorenz
Editorial Office Life Science Alliance
Meyerhofstr. 1
69117 Heidelberg, Germany
t +49 6221 8891 414

Reviewer #1 (Comments to the Authors (Required)):

The authors have addressed most of my comments to my satisfaction or provided convincing reasoning why a particular experiment would be beyond the scope of the manuscript. The manuscript is also considerably easier to read compared to the previous version.

My only remaining suggestion, which has not been addressed is, that I would prefer the text in the results section and the figure legends to mention clearly when an experiment has been carried out with bacterially produced protein. As in the previous version of the manuscript, the reader can figure this out, however, it takes time. This will be very very easy to address without negatively impacting the manuscript or the presentation of the work.

Few grammatical errors should also be corrected, for example Materials and Methods -> Preparation of recombinant proteins -> To prepare tag-free version, purified 2CPs and mutant 2CPBs were incubated with thrombin protease, and remove the His-tag -> this likely should be "to remove the His-tag". Other example: Section title "CYP20-3 block the peroxidase activity..." should be "CZP20-3 blocks the peroxidase activity...".

We are very pleased to submit our revised manuscript (#LSA-2020-00775-T, Liu et al.) for publication in your esteemed journal, *Life Science Alliance*. We wish to sincerely thank the editor and referees for very constructive suggestions. All these have been addressed in the revised manuscript. Please find our responses to the suggestions from referee #1 below. **A** = our answer/response to the referee's suggestions which are lettered in lowercase (e.g. a, b, c, etc).

Response to Referee #1

We have addressed the referee's comments and concerns regarding the sufficient introduction and discussion. Below we have described all the changes and modifications made in the revised manuscript as per the suggestions. **A** = our answer/response to the reviewer's suggestions/comments which are lettered in lowercase (e.g. a, b, c, etc).

a. I would prefer the text in the results section and the figure legends to mention clearly when an experiment has been carried out with bacterially produced protein.

A, As per the referee's suggestion, we have addressed that proteins used are recombinantly produced in bacteria (*E. coli*) in the revised result section and figure legends. These changes include;

- 1) "2CPs, **prepared recombinantly in *E. coli***, uniquely bind a negatively charged tripeptide GSH", (page #6 line #1 in the revised manuscript).
- 2) "In (A-D), recombinant 2CPs were **produced in *E. coli* and purified by a nickel-column**, as described in Materials and Methods", (the revised legend for Fig 1).
- 3) "The 1 μ M, tag-free **recombinant** 2CPA (A) and 2CPB (B)", (the revised legend for Fig 2).
- 4) "All proteins were tag-free, **recombinant versions prepared in *E. coli* BL21 (DE3)**", (the revised legend for Fig 3 and 4).
- 5) "2CPA and CYP20-3 were tag-free, **recombinant versions produced in *E. coli* BL21 (DE3)**", (the revised legend for Fig 5).
- 6) "The 2CPs were **recombinantly produced in *E. coli*, and purified by a nickel-column**, as described in Materials and Methods", (the revised legend for Fig S1 and S6).
- 7) "The 1.5 μ M **recombinant** 2CPs", (the revised legend for Fig S2).

- 7) "The 1.5 μ M **recombinant** 2CPs", (the revised legend for Fig S2).
 - 8) "The 2CPs were tag-free, **recombinant versions produced in *E. coli* BL21 (DE3)**", (the revised legend for Fig S3).
 - 9) "All proteins were tag-free versions, **recombinantly produced in *E. coli*** and purified by a nickel-column, as described in Materials and Methods", (the revised legend for Fig S7-S9).
- b. Few grammatical errors should also be corrected, for example Materials and Methods -> Preparation of recombinant proteins -> To prepare tag-free version, purified 2CPs and mutant 2CPBs were incubated with thrombin protease, and remove the His-tag -> this likely should be "to remove the His-tag". Other example: Section title "CYP20-3 block the peroxidase activity..." should be "CZP20-3 blocks the peroxidase activity..."
- A, As per the referee's suggestion, we have corrected the grammatical errors. These read **1)** "CYP20-3 **blocks** the peroxidase activity of 2CPA^{GS} during heat-shock stress" (page #10 line #3 in the revised manuscript), and **2)** "**to remove the His-tag**, purified 2CPs and mutant 2CPBs were incubated with thrombin protease" (page #14 line#22 in the revised manuscript).

Having addressed all of the concerns of the reviewers, we hope that the revised manuscript is now ready for publication in *Life Science Alliance*.

July 21, 2020

RE: Life Science Alliance Manuscript #LSA-2020-00775-TRR

Dr. Sang-Wook Park
Auburn Univeristy
Entomology and Plant Pathology
209 Rouse Life Science Bldg.
Auburn, Alabama 36849

Dear Dr. Park,

Thank you for submitting your Research Article entitled "CYP20-3 Deglutathionylates 2-CysPRX A and Suppresses Peroxide Detoxification during Heat Stress". It is a pleasure to let you know that your manuscript is now accepted for publication in Life Science Alliance. Congratulations on this interesting work.

DISTRIBUTION OF MATERIALS:

Again, congratulations on a very nice paper. I hope you found the review process to be constructive and are pleased with how the manuscript was handled editorially. We look forward to future exciting submissions from your lab.

Sincerely,

Reilly Lorenz
Editorial Office Life Science Alliance
Meyerhofstr. 1
69117 Heidelberg, Germany
t +49 6221 8891 414
e contact@life-science-alliance.org
www.life-science-alliance.org